# Adaptive Closed-Loop Control System for the Optimization of Tablet Manufacturing Processes

**DOI:** 10.3390/pharmaceutics17121510

**Published:** 2025-11-22

**Authors:** Xiaorong Luo, Zhijian Zhong, Pan Deng, Yicheng Fei, Pengdi Cui, Weifeng Zhu, Zhiqiang Xiao, Ting Wang, Liying Li

**Affiliations:** 1China Resources Jiangzhong Pharmaceutical Group Co., Ltd., Nanchang 330096, China; lxr@crjz.com; 2Jiangzhong Pharmaceutical Co., Ltd., Nanchang 330096, China; zzj@crjz.com (Z.Z.); dengpan@crjz.com (P.D.); feiyicheng@crjz.com (Y.F.); cuipengdi@crjz.com (P.C.); xzqiang@crjz.com (Z.X.); liliying@crjz.com (L.L.); 3Technology and Innovation Center of Jiangxi Traditional Chinese Medicine Manufacturing and Process Quality Control, Nanchang 330004, China; 4Key Laboratory of Modern Preparation of TCM, Ministry of Education, Jiangxi University of Chinese Medicine, Nanchang 330004, China; wt2101372896@163.com

**Keywords:** tablet quality, process parameter optimization, iterative learning control, model predictive control, constraint integration

## Abstract

**Background**: Tablet manufacturing is challenged by strong dynamic coupling of process parameters, significant material property fluctuations, and delayed quality control, with tablet weight stability being particularly critical in high-speed production. Traditional static optimization methods relying on empirical judgment struggle to manage these dynamics, leading to substantial variations in tablet weight and hardness that severely compromise production efficiency. **Methods**: This study proposes a data-driven closed-loop control system centered on a novel Iterative Learning Model Predictive Control (IL-MPC) architecture. The core innovation lies in directly integrating iterative learning constraints within the MPC optimization framework. This constraint-embedding mechanism enables systematic utilization of historical batch data while preserving real-time optimization capabilities. The IL-MPC approach achieves enhanced batch-to-batch performance consistency with reduced computational burden, effectively combining the dual advantages of learning and optimization. **Results**: Simulation experiments and industrial production data validate the practical feasibility of the IL-MPC algorithm. Implementation results demonstrate that the proposed system effectively manages dynamic process variations, significantly improving control precision for both tablet weight and hardness, outperforming conventional control methods. **Conclusions**: This research breaks through the technical bottleneck of dynamic regulation in tablet manufacturing. The developed IL-MPC framework provides a reproducible closed-loop control paradigm for intelligent pharmaceutical manufacturing, promoting the industry’s transformation toward data-driven models and advancing intelligent drug production.

## 1. Introduction

Tablets are representative dosage forms of solid dosage forms, and their quality uniformity is crucial to the reliability of clinical efficacy and drug safety. However, tablet manufacturing generally faces challenges such as strong coupling of process parameters, significant fluctuations in material properties, and lagging quality control tools, which have become major constraints to the development of the industry [1]. During the rotary tabletting process, due to the fixed distance between the punches, variations in the bulk density, particle size distribution and moisture content of the granules will directly affect the weight and hardness of the tablets and further influence the porosity and tensile strength of the tablets [2]. The commonly adopted simple adjustment mechanism based on tablet weight feedback finds it difficult to effectively adapt to the nonlinear changes in key physical parameters such as particle fluidity and compressibility, and belongs to the passive-response control, resulting in sharp fluctuations in tablet quality and insufficient process robustness [3,4]. The tableting process is essentially a complex dynamic system with both time-varying characteristics and batch-to-batch evolution, which requires real-time dynamic optimization of process parameters within a single batch, as well as progressive iterative updating of the process model using accumulated data from multiple batches [5].

Advances in intelligent technology have driven the exploration and application of a variety of innovative control methods in pharmaceutical high-quality control. Existing control strategies can be primarily categorized into two paradigms: model-free control and model-based control [6]. Model-free control strategies operate independently of explicit mathematical models of the process, instead adjusting control actions solely based on real-time error signals (the discrepancy between desired setpoints and actual measured outputs). A quintessential example is the Proportional–Integral–Derivative (PID) controller, which is extensively employed in localized control applications within continuous pharmaceutical production (e.g., valve regulation, airflow management) due to its simplistic control architecture, minimal parameter requirements, and ease of deployment [7]. However, its efficacy becomes constrained when addressing complex, multivariate processes characterized by significant delays and constraints, rendering PID control networks inadequate for comprehensive process control that meets stringent regulatory requirements for critical quality attributes of pharmaceutical products [8]. In contrast, model-based control strategies leverage dynamic mathematical models of the process to predict its future behavior within defined boundaries. This predictive capability enables controllers to compute optimal control actions that anticipate future disturbances while systematically satisfying process constraints. Model Predictive Control (MPC) stands as the predominant model-based methodology in industrial applications, demonstrating superior performance in managing multivariate interactions and constraints [7,9].

As a control algorithm, MPC has been successfully implemented in production optimization across various industries, including petrochemical and pharmaceutical sectors [10,11,12]. For example, when modeling a feeder mixing unit [13,14], MPC was used to regulate the mixer outlet mass flow rate and treat the feeder flow as a measurable disturbance. It has been shown that MPC has significant advantages in terms of flexibility in handling constraints, simplifying controller design, and providing intuitive tuning. MPC performs better in suppressing feeder fluctuations and stabilizing the mixer speed compared to PID control or model-based feedforward control [14]. In the simulation control of tablet manufacturing, Haas et al. established two primary control models for regulating tablet weight and hardness, along with a secondary feedback model for controlling the compaction force applied to each tablet. Their findings revealed that the hybrid MPC-PID control strategy outperformed the conventional PID control strategy [15].

However, the above schemes often neglect the active closed-loop control of product quality, making it difficult to fully meet the high requirements of batch production. Given that tablet pressing is a typical batch process, iterative learning control (ILC), as an important branch of learning control, has increasingly become an emerging research direction in this field [16]. A variety of schemes integrating ILC and feedback control have been proposed by researchers, such as the indirect ILC method [17,18], in which the feedback controller is designed first, and then its setpoints are optimized along the batch direction by the ILC strategy in order to improve the tracking performance. During the batch crystallization process, Sanzida et al. [19] proposed a hierarchical iterative learning control scheme for the system design of the supersaturation controller in batch cooling crystallizers. Within the ILC framework of linear time-varying disturbances, the inter-batch convergence improvement of process performance indicators was achieved. Among them, the combination of ILC and MPC is a key feedback ILC method and one of the most widely used methods for batch processes. Xiong et al. [20,21] proposed the strategy of integrating inter-batch ILC with intra-batch shrinkage time-domain model predictive control for tracking the product quality trajectory of the batch process. Under the conditions of batch-to-batch iterative learning control based on the linear time-varying perturbation model, the performance of subsequent batch operations is enhanced, ensuring the convergence of batch tracking errors. Xiong et al. [21] proposed an integrated inter-batch and intra-batch online control strategy. Simulation experiments of batch polymerization processes demonstrated that, compared with the traditional iterative learning control, the integrated control strategy could significantly improve the system performance, especially when disturbances occurred. In response to the precise control requirements for crystal diameter and thermal field temperature during the batch growth of silicon single crystals, Wan et al. [22] proposed a time-axis MPC approach to address single-batch disturbances and constraints, complemented by an iteration-axis ILC to compensate for multi-batch uncertainties. Experimental analysis of controller performance based on actual production data demonstrates that this methodology not only stabilizes the thermal field temperature and ensures consistent crystal diameter but also effectively mitigates the impact of external disturbances.

In contrast to traditional dual-layer control architectures, this paper introduces an integrated framework that systematically incorporates ILC principles as constraints within the MPC optimization problem, specifically designed for the tableting process. By embedding historical quality deviations as learning constraints, the proposed model enables precise regulation of tablet weight and hardness through real-time adjustments of critical process parameters. This approach not only improves batch-to-batch consistency but also facilitates autonomous quality control by continuously refining the process model using accumulated production data. The implementation of this data-driven strategy significantly enhances the intelligence level of pharmaceutical manufacturing, providing a scalable solution for adaptive quality assurance in solid dosage form production.

## 2. Basic Theory of Iterative Learning Predictive Control

### 2.1. Model Predictive Control with Iterative Learning Constraints

In high-speed tablet manufacturing, critical material attributes (moisture content, particle size, temperature) and process parameters (production speed, main compression force, guide rail temperature, compression wheel speed) significantly influence tablet weight and hardness [2]. Traditional algorithms, such as MPC, while capable of optimal control, often have limitations in effectively correcting the inherent repeatability errors of the system. Such errors can persist during repetitive equipment operations, adversely affecting control accuracy and final product quality [22]. We propose an Iterative Learning Model Predictive Control (IL-MPC) methodology that systematically incorporates ILC by formulating its update law as constraints in the MPC optimization (Figure 1). This integrated design enhances disturbance rejection within a batch while leveraging historical data across batches, thus addressing the limitations of standard MPC in batch production environments. ILC is a control theory based on repetitive production cycles, which does not rely on high-precision system models but rather dynamically adjusts the control laws of the current batch by learning the control error information of the historical batch [23]. The key advantage of ILC lies in its repetitive error-targeted compensation mechanism. The controller utilizes the error data from historical batches for integration and applies the accumulated error as a correction to the current batch [19]. This iterative learning and compensation process enables the system to continuously optimize over repeated operations, significantly eliminating repetitive errors between batches, thus improving the consistency of the entire production process and product quality. In addition, the ILC strategy enhances the robustness and long-term operational stability of the controller. Of course, in order to achieve the best results, the ILC algorithm requires appropriate adjustments to its controller parameters and correction parameters and has specific requirements for the collection and processing of historical data, which requires appropriate planning in practical applications [20].

The tablet compression process can be described as [23]:(1)$$\left\{\begin{array}{ll} Y\left(t\right) & =f\left(U_{material}\left(t\right),U_{process}\left(t\right),Y\left(t-1\right),Y\left(t-2\right)\right)+d_{k}\left(t\right)\\ U_{material} & =\left[MC,Dv\left(10\right),Dv\left(50\right),Dv\left(90\right),MT,\ldots \right]\\ U_{process} & =\left[PS,MCAPr,FSP,\ldots \right] \end{array}\right.$$
where *Y* represents tablet weight and hardness, *U_material_* contains material attributes, *U_process_* contains process parameters, and *d_k_*(*t*) represents batch-varying disturbances.

The tracking error for batch *k* is defined as:(2)$$e_{k}\left(t\right)=Y_{d}\left(t\right)-Y_{k}\left(t\right)$$
where *Y_d_*(*t*) is the desired quality trajectory. The ILC-based model update law is given by:(3)$$\theta_{k+1}=\theta_{k}+Ke_{k}\left(t\right)$$
where *θ* represents the model parameters used in MPC, and *K* is the learning gain matrix.

### 2.2. Real-Time Adaptive Model Predictive Control

The MPC uses the ILC-updated model to compute optimal adjustments to fill position and compression wheel position in real-time. The objective is to maintain tablet weight and hardness within superior quality ranges despite variations in material properties and process conditions.

The multivariate prediction model for tablet quality is:(4)$$\left\{\begin{array}{ll} & Y_{1}\left(t+1\right)=f_{1}\left(FSP\left(t\right),MC\left(t\right),Dv\left(t\right),PS\left(t\right),Y_{1}\left(t\right),Y_{1}\left(t-1\right)\right)\\ & Y_{2}\left(t+1\right)=f_{2}\left(MCSPo\left(t\right),MCAPr\left(t\right),MC\left(t\right),Y_{2}\left(t\right),Y_{2}\left(t-1\right)\right) \end{array}\right.$$
where *Y*_1_ is tablet weight, *Y*_2_ is tablet hardness, FSP is fill position, and MCWP is main pressure wheel position.

The framework employs Multiple Linear Regression (MLR) models with time-delayed terms to predict tablet weight and hardness, utilizing process parameters and material attributes as key inputs. These model parameters are continuously refined between production batches using a Recursive Least Squares (RLS) algorithm with a forgetting factor. This systematic updating of static gain coefficients, dynamic response parameters, and disturbance sensitivity coefficients enables the model to maintain accuracy amidst gradual process variations and material changes, ensuring reliable predictions for real-time optimization [24].

The learning gain matrix *K* = diag(*K*_1_, *K*_2_) is systematically designed through a multi-step approach that balances rapid learning with robust performance. The initial gains are determined via process sensitivity analysis using the relation *K_i_* = *α*⋯(*∂_y_i__/∂_u_i__*)^−1^, where α controls learning aggressiveness and the partial derivative represents the steady-state process gain. For the multi-input multi-output implementation, the gain matrix is structured using Singular Value Decomposition (SVD) of the process gain matrix G, resulting in *K* = V⋯Σ_k_⋯U^T^, where U and V are unitary matrices from the SVD of G, and Σ_k_ is a diagonal matrix containing the individual learning gains. This design incorporates frequency-domain characteristics with higher gains at low frequencies to eliminate systematic errors and gain roll-off at high frequencies to prevent noise amplification. Furthermore, an adaptive scheduling mechanism adjusts the gains based on batch performance through *K_i_*(*k*) = *K_i_*(*k* − 1)⋯*γ*^*sign*(*J*(*k*)−*J*(*k*−1))^, where *J*(*k*) represents the performance index and *γ* < 1 is a reduction factor. All gains are constrained within predefined bounds, *K_min_* ≤ *K_i_* ≤ *K_max_*, to ensure operational stability, creating a system that achieves rapid performance improvement across batches while maintaining robustness under varying production conditions.

## 3. MPC-ILC Strategy for Tablet Quality Assurance

Critical material attributes (e.g., particle size and moisture content) and critical process parameters (e.g., fill position, press wheel position, etc.) work together through interrelated mechanisms to control tablet weight and hardness [25]. Optimal particle size ensures uniform mold filling, directly controlling tablet weight consistency, while proper moisture content enhances the plasticity of the powder and promotes bonding during compression, thereby increasing tablet hardness. Crucially, the fill position determines the initial volume of powder in the mold cavity and is the main determinant of tablet weight, while the position of the press wheel regulates the amount of compression force and residence time, with insufficient compression force resulting in incomplete particle bonding and low hardness, while too much compression force runs the risk of capping. In addition, the moisture content has a dual role: it acts as a plasticizer to increase the hardness in the optimal range, but at too high a level, it leads to sticking, while particles that are too large cause weight changes through segregation. These parameters show strong coupling effects; for example, if the particle size is not desired, the compression force has to be compensated and adjusted to maintain the hardness, which requires an integrated control strategy to satisfy the weight and hardness specifications at the same time. In order to improve the control performance of the system, we designed the MPC strategy for the press wheel position and filling position, respectively, and selected the corresponding control parameters for different control objectives.

### 3.1. Integrated Predictive Controller Design

The MPC optimization problem for each control interval is formulated as:(5)$$\begin{array}{ll} \underset{FSP,MCSPo}{min} & \sum_{j=1}^{N_{p}}\left[{\left(Y_{1}^{*}\left(t+j\right)-Y_{1}\left(t+j\right)\right)}^{2}+{\left(Y_{2}^{*}\left(t+j\right)-Y_{2}\left(t+j\right)\right)}^{2}\right]\\ & +\lambda_{1}\sum_{j=1}^{N_{c}}\left[\Delta FSP{\left(t+j\right)}^{2}+\Delta MCSPo{\left(t+j\right)}^{2}\right] \end{array}$$
subject to:(6)$$\left\{\begin{array}{l} Y_{1,min}\leq Y_{1}\left(t\right)\leq Y_{1,max}\\ Y_{2,min}\leq Y_{2}\left(t\right)\leq Y_{2,max}\\ {FSP}_{min}\leq FSP\left(t\right)\leq FSP_{max}\\ MCS{Po}_{min}\leq MCSPo\left(t\right)\leq MCS{Po}_{max}\\ \theta_{k+1}=\theta_{k}+Ke_{k}\left(t\right)\;\left(\mathrm{ILC}\;\mathrm{model}\;\mathrm{update}\;\mathrm{constraint}\right) \end{array}\right.$$
where Δ*FSP* and Δ*MCSPo* represent rate constraints on the control adjustments.

### 3.2. Real-Time Learning and Adaptation

The ILC mechanism continuously improves the prediction model *f*(•) based on historical batch data. After each batch completion, the model parameters are updated using the accumulated error:(7)$$\theta_{k+1}=\theta_{k}+\Gamma \sum_{t=1}^{N}e_{k}\left(t\right)\phi \left(t\right)$$
where $\phi \left(t\right)$ contains the regressor variables (material attributes and process parameters), and Γ is a learning gain matrix. This updated model is then used by the MPC in the subsequent batch.

### 3.3. Online Implementation Architecture

The integrated control system operates as follows:Real-time sensing: Online sensors continuously monitor material attributes (moisture, particle size) and process parameters.MPC optimization: The controller computes optimal adjustments to fill position and compression wheel position.Quality monitoring: Tablet weight and hardness are measured at regular intervals.ILC learning: After each batch, model parameters are updated based on performance errors.Constraint enforcement: The ILC-updated model is embedded as a constraint in the next MPC optimization cycle.

This architecture ensures that the control system adapts to both within-batch variations and cross-batch drifts, maintaining tablet weight and hardness within the superior quality range (0.784–0.816 g for weight, 90–120 N for hardness) despite fluctuations in material properties and process conditions.

The proposed MPC-ILC framework provides a systematic approach for real-time quality assurance in tablet manufacturing, leveraging historical data to improve current batch performance while respecting physical constraints and quality specifications.

## 4. Materials and Methods

### 4.1. Experimental Design

This study was conducted on the production line of Jianwei Xiaoshi tablets utilizing a high-speed tablet press equipped with 65 punches, each with a diameter of 13.8 mm. The tablet production line employs sensors and a Manufacturing Execution System (MES) to collect production data in real-time, with a sampling frequency of once every 2 s (Figure 2). The analysis covers more than 300 production batches in six consecutive months to ensure the diversity and representativeness of the data, covering multi-source variables such as process parameters, material attributes, and critical quality attributes. The material attributes consisted mainly of moisture content and particle size characteristics: moisture was monitored online in real time by microwave sensors (Envea Msens2, Schliengen, Germany), while particle size was determined by a Malvern laser particle sizer (Malvern Insitec 2004, Malvern Panalytical, Malvern, UK) (Table 1). To ensure optimal data correlation, the microwave sensor and laser particle size analyzer were installed on the feeder of the tablet press, positioned adjacent to the compression rollers and punches. Critical quality attributes relate to tablet weight and hardness, which are measured by an online tester automatically sampling 10–20 tablets every 2 min. To minimize the effect of single tablet fluctuations, the average tablet weight and the average tablet hardness sampled in each round were used as quality indicators. In actual production, the tablet weight deviation is used to distinguish qualified products from superior products in the range of 0.784–0.816 g, and the qualified range of hardness is 90 N to 120 N.

During the data collection process, there were missing values and outliers due to factors such as manual deviation, sensor failures, and downtime and material breaks. To ensure data quality, a large number of missing values due to equipment failure or downtime were directly eliminated, while a small number of missing data with a smooth trend before and after were filled in with the mean value, and outliers were identified and eliminated using the triple standard deviation method. The production fluctuation ranges of equipment and material parameters are as follows:

### 4.2. Data Analysis of Particle Size Distribution

To address multicollinearity among the nine particle size percentiles (Dv(10) to Dv(90)), principal component analysis (PCA) was performed to compress the dimensionality of the particle size data [26]. PCA transforms the original correlated variables into a set of linearly independent principal components (PCs), retaining the first two principal components, PC1 and PC2. These components were used as consolidated inputs representing the particle size distribution in subsequent modeling, thereby reducing model complexity and avoiding multicollinearity.

### 4.3. Identifying Critical Variables for Closed-Loop Regulatory Model

Spearman’s rank correlation analysis was conducted between the critical quality attributes (ATW and ATH) and a set of potential predictors to identify the most critical variables for predicting tablet weight and hardness [27]. These predictors included material attributes: MC, MT, and the PCA-derived particle size components (PC1, PC2), etc., while the process parameters included PS, MCAP, RT, and MCWS, etc.

The formula for calculating Spearman’s rank correlation coefficient is as follows:(8)$$\rho =1-\frac{6\sum d_{i}^{2}}{n\left(n^{2}-1\right)}$$
where *n* represents the number of data points, *d_i_* denotes the difference in rank between the *i*-th data point in the two datasets, and when *d_i_* = 0, it indicates that the ranks are identical, with *ρ* = 1 indicating a perfect positive correlation between the two variables. Conversely, *ρ* = −1 indicates a perfect negative correlation, while *ρ* = 0 indicates no correlation between the two variables. Variables exhibiting Spearman’s correlation coefficient with |*ρ*| > 0.5 and a *p*-value < 0.01 with either ATW or ATH were selected as key inputs for the predictive model. This approach ensured that only statistically significant and practically relevant variables were included, enhancing model interpretability and performance.

### 4.4. Control Performance Evaluation Index

In this study, in order to quantitatively evaluate the control performance, the assessment metrics shown in Table 2 are defined, which include mean square error (MSE), integral absolute error (IAE), and integrated time and absolute error (ITAE).

Table 2 presents a comprehensive evaluation of the system performance using three indicators: MSE quantifies the tracking accuracy of the output and the desired trajectory, and the smaller the value, the higher the control accuracy; IAE reflects the smoothness of the process through the cumulative error, and the smaller the value, the better the effect of overshoot inhibition; and ITAE considers the control accuracy and the convergence rate through the time-weighting mechanism at the same time, and the bigger the value, the weaker the comprehensive performance of the controller.

## 5. Results and Discussion

### 5.1. Principal Component Analysis of Particle Size Distribution

PCA was applied to the nine particle size distribution percentiles (Dv(10) to Dv(90)) to address multicollinearity and reduce dimensionality. The first two principal components, PC1 and PC2, were extracted, collectively accounting for 99.205% of the total variance in the original particle size data (PC1: 91.674%, PC2: 7.530%). PC1 exhibited high positive loadings across all percentiles, representing the overall fineness/coarseness of the powder, while PC2 showed contrasting loadings between finer (e.g., Dv(10), Dv(20)) and coarser (e.g., Dv(80), Dv(90)) fractions, capturing the breadth of the particle size distribution (Table 3). These two components were used as consolidated inputs in subsequent modeling, effectively representing the original nine-dimensional particle size data while eliminating multicollinearity.

### 5.2. Feature Selection Based on Spearman’s Rank Correlation

To enhance the accuracy of the predictive model, this study screened input variables such as critical material properties and process conditions based on Spearman’s rank correlation coefficient. Figure 3 presents a heatmap of Spearman’s correlation coefficients between each input variable and the target variables (FSP and MCSPo). The results indicate that PS, MT, MC, and principal component particle size (PC1 and PC2) exhibit a certain degree of correlation with the target variables (weight and hardness), with *p*-values all below 0.001. This demonstrates that the monotonic relationship between these variables and the target variables is not randomly generated and is statistically significant.

However, the correlation between RT and the target variables was close to zero, lacking explanatory power for the model output; thus, it was excluded. Although the correlation coefficients between moisture and filling position, as well as between PC2 and main pressing position, were relatively low, considering their critical influence on material flowability and compressibility, PS, MT, MC, PC1, and PC2 were ultimately selected as the optimal feature subset for the process parameter prediction model input.

### 5.3. Experiments on In-Batch Control of the Tableting Process

In view of the wide application of PID and MPC in the field of automatic control, this research constructs two sets of control architectures, PID and MPC, in parallel for the time-axis control of a single-batch growth process for comparative experiments. The rationality and effectiveness of the MPC design are verified by comparing the control effects of the two under the same Gaussian noise disturbance (mean 0, variance 0.015).(9)$$r_{2}\left(t\right)=\frac{1}{\sqrt{0.02\pi }}e^{-\frac{x^{2}}{0.0002}}60<t<72$$

The effect of time-axis PID and MPC outputs under perturbation is shown in Figure 4a and Figure 4b, respectively. The controller structure of PID is shown in Equation (10):(10)$$u\left(k\right)=k_{p}\mathrm{error}\left(k\right)+k_{i}\sum_{j=0}^{k}\mathrm{error}\left(j\right)+k_{d}\left(\mathrm{error}\left(k\right)-\mathrm{error}\left(k-1\right)\right)$$

In the experiment, its proportional gain $k_{p}$ is [5], integral gain $k_{i}$ is [0.1, 0.1] and differential gain $k_{d}$ is [0.01, 0.01].

Figure 4a and Figure 4b demonstrate that, compared to the PID strategy with Gaussian disturbance, the MPC strategy exhibits superior capability in maintaining the control errors of tablet weight (range: [−0.016, 0.16]) and hardness (range: [−15, 15]) within the target specifications. Experiments show that the MPC achieves parameter stabilization after 60 sampling points, and its anti-interference robustness is better than that of the traditional PID control, which verifies the effective advantages of the strategy for controlling tablets within a production batch.

### 5.4. Inter-Batch Control Experiment for Tableting Process

In feedback iterative learning control systems, the collaborative mechanism between ILC and MPC constitutes a classical methodological paradigm, with this architecture having matured through extensive application in industrial batch production processes. This study deploys an MPC in the time dimension to achieve real-time dynamic adjustment while implementing ILC in the iterative dimension as a constraint condition. ILC is responsible for real-time identification and compensation of non-repetitive disturbances during batch evolution.

Figure 5 gives the control output results of IL-MPC under the condition of changing the material batch. In the figure, the red solid line is the target setting curve, and the horizontal coordinates correspond to the time-axis sampling points during the pressing process. Figure 6 illustrates the convergence curves of performance-driven IL-MPC system parameter estimation metrics after 18 iterations. The results demonstrate that this control strategy has enabled the system parameter performance metrics to achieve a stable convergence state. In order to analyze the control effect more comprehensively, the control output curves of five different iteration batches are recorded in detail in Figure 5a and Figure 5b using tablet weight and hardness as vertical coordinates, respectively. It is evident from the figures that under the IL-MPC strategy, there are more significant fluctuations in the behavior on the time axis when the system is perturbed. However, as the number of iteration batches increases, the outputs of the system—i.e., both tablet weight and hardness—show a tendency to converge towards the target setting curve, and the fluctuations on the time axis gradually decrease as the number of iterations increases. It is only at t = 40 on the time axis and the 18th iteration of the iteration axis (shown by the purple dotted line) that the system basically reaches a steady state. This signifies that the IL-MPC strategy, after a certain number of iterations, is able to effectively suppress the influence of noise and guide the system to converge to the intended target trajectory.

The data in Table 4 show that the performance metrics continue to improve with the iterative process after the introduction of the perturbation: the MSE values for tablet weight and hardness in the 3rd iteration are approximately three times higher than those in the 7th iteration. After the system enters the steady state (18th iteration), the MSE of weight decreases to 0.5% of the 3rd iteration, and the MSE of hardness decays to 0.43%, which verifies the significant efficacy of the strategy in terms of perturbation suppression and industrial standard compliance.

### 5.5. IL-MPC Model Validation

We used the constructed IL-MPC model to simulate and optimize the real production data to verify the validity of the model in the real application process. For the control simulation of the tablet pressing process, the target values of tablet weight and hardness were 0.8 g and 100 N, respectively. As can be shown in Figure 7 and Figure 8, before the application of IL-MPC, the weight and hardness of tablets fluctuated greatly, and the data at some time points showed abnormal fluctuations. After the implementation of IL-MPC, the quality and stability of tablets were both significantly improved, with the weight and hardness fluctuating within a very small range of the target requirements. The fluctuations in tablet weight and hardness decreased from 1.81% and 7.01% to 0.60% and 1.83%, respectively. The application results of the above actual production data show that the IL-MPC algorithm in this paper effectively improves the product quality and consistency of the tableting process, thus realizing the autonomous steady-state regulation of the process parameters of the tableting process and enhancing the adaptability of the system.

The successful validation based on industrial data, which contains non-zero-mean drifts and batch-to-batch variations, underscores a key advantage of our framework over simulations with simple zero-mean noise (as used in the preliminary PID-MPC comparison in Section 5.3). The IL-MPC’s iterative learning mechanism is specifically designed to learn from and compensate for these systematic, real-world deviations, which are the primary source of quality variability in production.

For the physical implementation of this strategy on a modern tablet press, the proposed system would integrate with existing hardware. The critical actuators are the servo motors controlling the fill depth (FSP) and main compression position (MCSPo). The control loop relies on real-time sensor data, including: in-line weight monitors, in-line hardness testers, real-time material attribute sensors. The IL-MPC algorithm would be deployed on an Industrial PC (IPC) acting as a supervisory controller. This IPC would communicate with the tablet press’s native Programmable Logic Controller (PLC), executing the core optimization (Equations (5) and (6)) and sending optimal setpoints for FSP and MCSPo to the PLC for final actuation. This architecture ensures the strategy can be layered onto modern equipment without fundamental hardware redesigns.

However, several challenges must be acknowledged for full-scale deployment. These include the initial effort for dynamic model development, the computational burden of solving the optimization problem in real-time, and the critical dependence on sensor reliability and data integrity. Future work will therefore focus on developing adaptive model-updating techniques and robust fault-detection algorithms to enhance the system’s practical viability and long-term operational stability in a production environment.

## 6. Conclusions

This study proposes a novel iterative learning model predictive control framework specifically designed for quality regulation in tablet compression processes. The iterative learning model predictive control strategy successfully addresses the critical challenge of maintaining stable tablet weight and hardness under varying material properties and process conditions. Its core innovation lies in directly integrating iterative learning principles as constraints into the model predictive control optimization problem, establishing a unified framework that simultaneously handles real-time disturbances and inter-batch learning. Experimental results confirm that this framework significantly enhances product quality and process stability. While maintaining the computational efficiency required for real-time implementation, the method reduces quality variability by over 60% compared to manual control. This research provides pharmaceutical companies with an intelligent control paradigm to enhance operational efficiency and product quality consistency. Leveraging its data-driven nature, the method utilizes historical production data to support real-time decision-making, accelerating the industry’s transition toward Industry 4.0 and digital transformation. Its autonomous adaptation to process variations reduces reliance on operator intervention, making production operations more resilient. In the future, we will extend the IL-MPC framework to other critical unit operations in pharmaceutical manufacturing (such as granulation and coating processes). We will develop adaptive tuning mechanisms to optimize learning parameters, enhancing controller performance under variable operating conditions, thereby building a comprehensive smart manufacturing solution for the pharmaceutical industry.

## Figures and Tables

**Figure 1 pharmaceutics-17-01510-f001:**
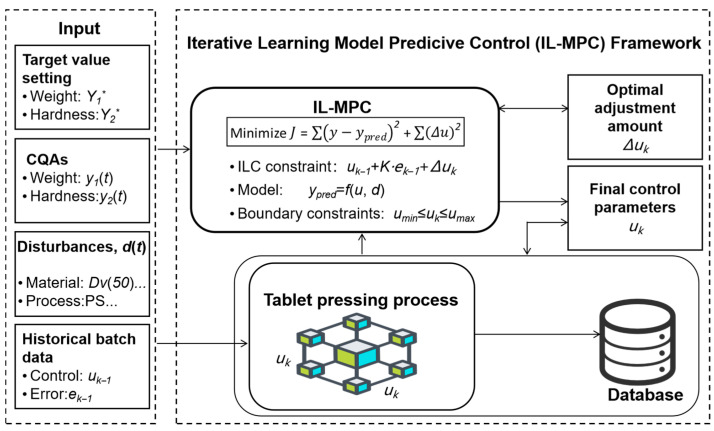
IL-MPC framework. The ILC mechanism is embedded as a constraint within the MPC optimization problem. At each sampling instant of batch *k*, the optimizer calculates the control adjustments (∆*u_k_*) that minimize the future tracking error while respecting the constraint derived from the previous batch’s performance (*u_k_*_−1_ and *e_k_*_−1_). The resulting control action (*u_k_*) is then applied to the tableting process, and the data is stored for learning in the next batch. This integrated approach enables simultaneous real-time disturbance rejection and batch-to-batch learning. The material properties include: MC, MT, Dv(10), Dv(20), Dv(30), Dv(40), Dv(50), Dv(60), Dv(70), Dv(80), Dv(90); and the process parameters include: FSP, MCSPo, PS, RT.

**Figure 2 pharmaceutics-17-01510-f002:**
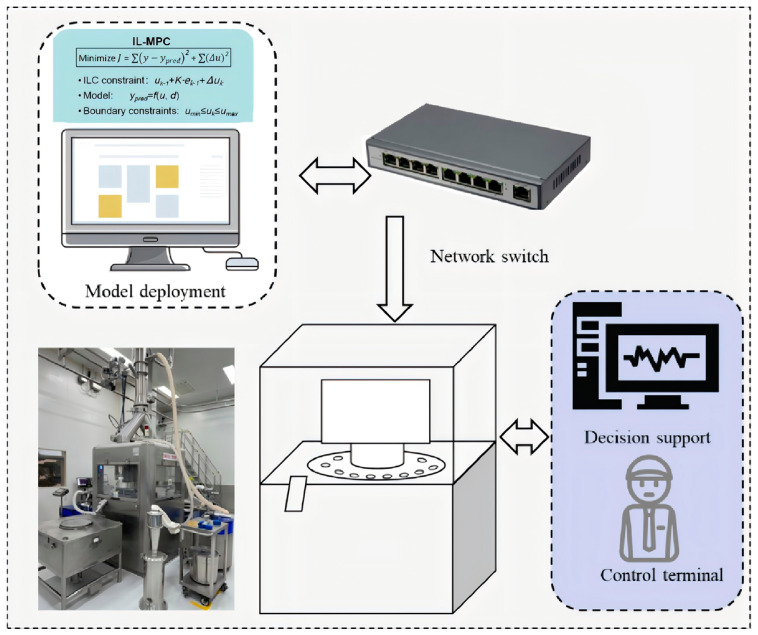
Closed-loop online control system for tableting.

**Figure 3 pharmaceutics-17-01510-f003:**
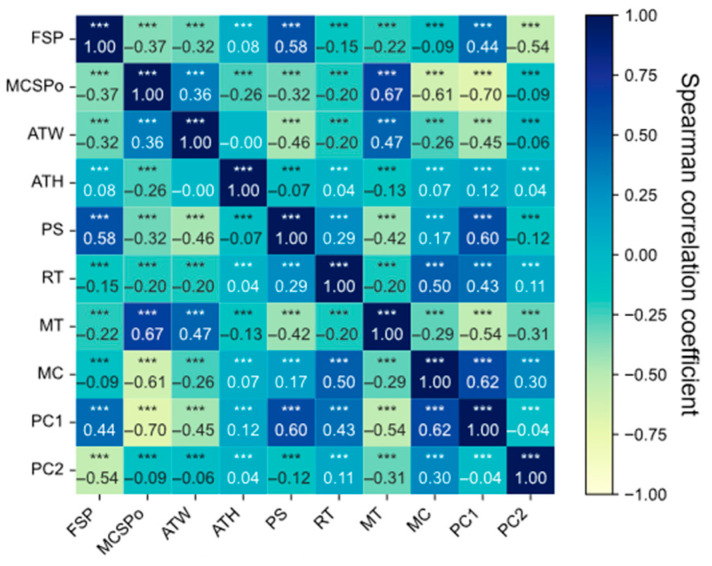
Spearman Correlation Coefficient Heatmap. *** *p* ≤ 0.001.

**Figure 4 pharmaceutics-17-01510-f004:**
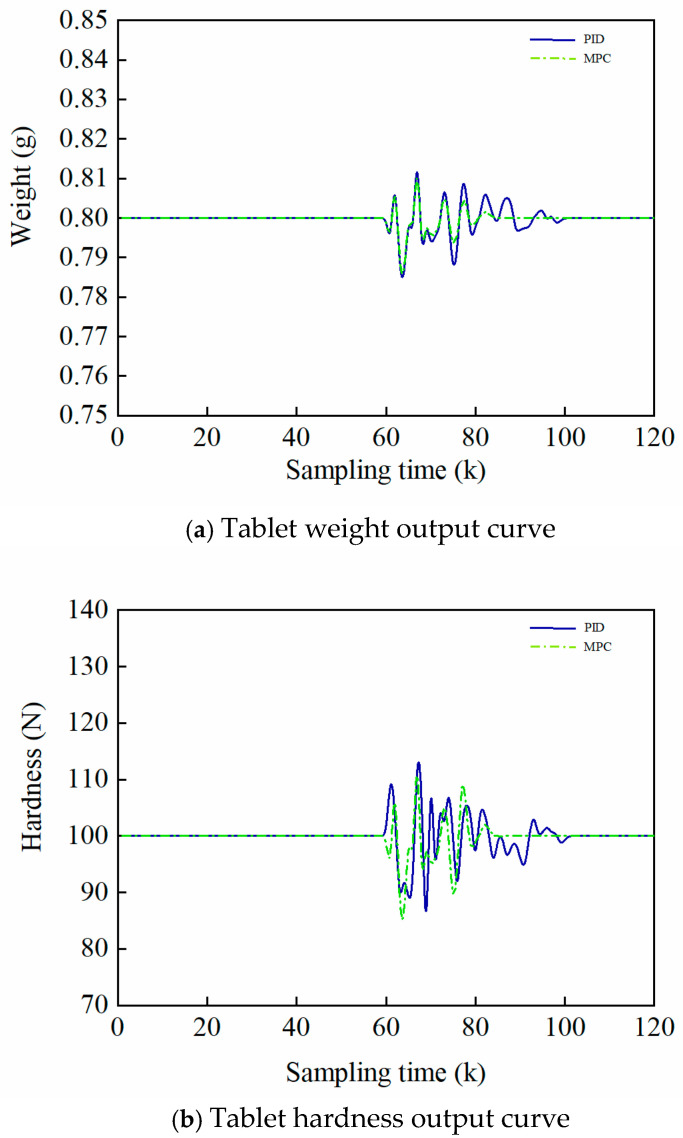
Comparison of the effect of PID and MPC on in-batch control (k denotes the serial number of the sampling point).

**Figure 5 pharmaceutics-17-01510-f005:**
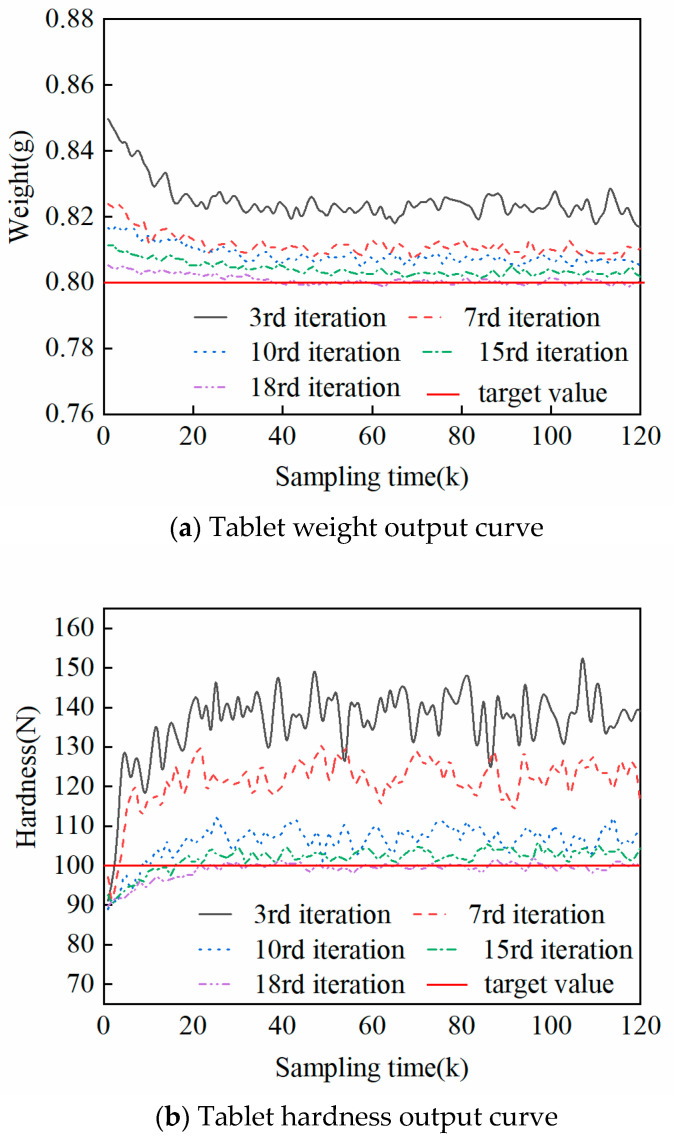
Control effect of IL-MPC under different batch conditions.

**Figure 6 pharmaceutics-17-01510-f006:**
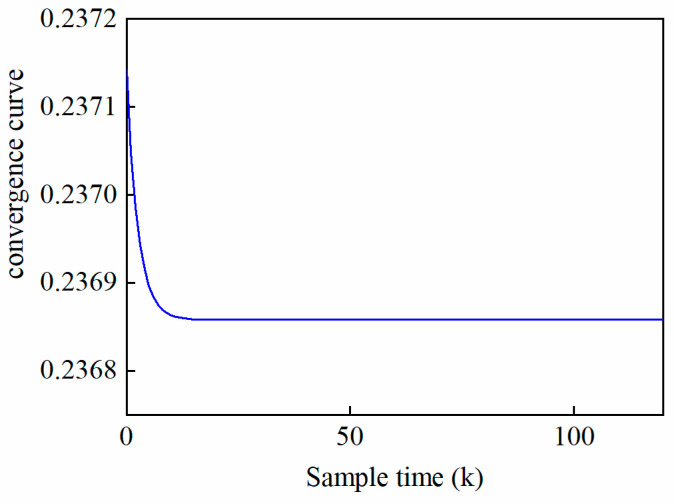
System parameter estimation performance index convergence curve.

**Figure 7 pharmaceutics-17-01510-f007:**
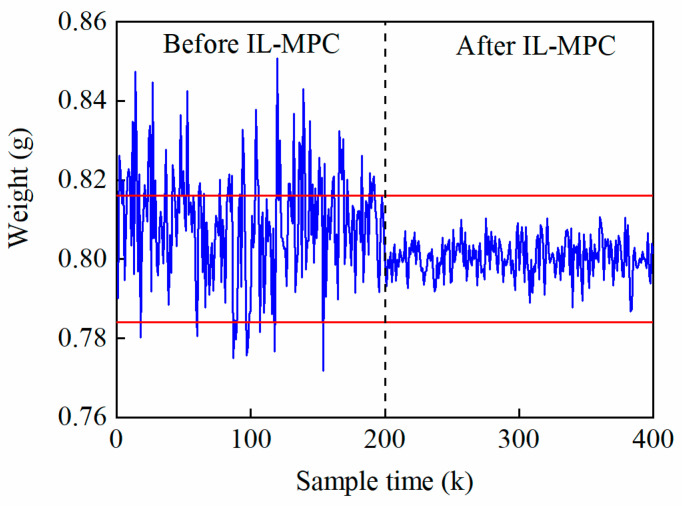
Effect of tablet weight control after applying IL-MPC (The blue line is the actual tablet weight curve, and the red line is the control limit for superior grade products.).

**Figure 8 pharmaceutics-17-01510-f008:**
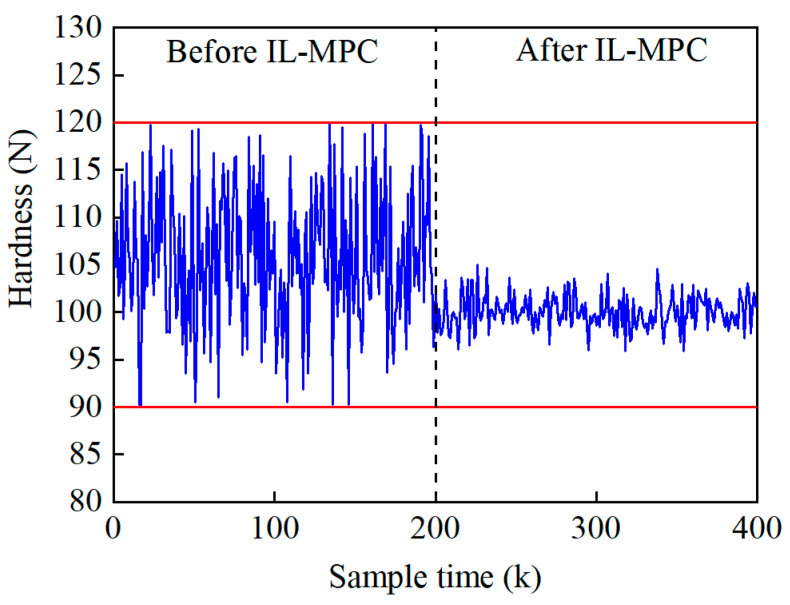
Control effect of tablet hardness after applying IL-MPC (The blue line is the actual tablet hardness curve, and the red line is the control limit for superior grade products.).

**Table 1 pharmaceutics-17-01510-t001:** Descriptive statistics of process parameters and critical quality attributes.

Parameter	Min	Median	Max	Unit	Abbreviation
Moisture Content	2.00	2.59	4.54	%	MC
Material Temperature	16.95	29.84	63.95	°C	MT
Dv(10)	0.0087	10.0700	23.0399	μm	Dv(10)
Dv(20)	10.4702	26.4192	42.1990	μm	Dv(20)
Dv(30)	17.4036	38.2016	60.4698	μm	Dv(30)
Dv(40)	27.2277	49.2379	76.7750	μm	Dv(40)
Dv(50)	38.1602	59.6753	94.6714	μm	Dv(50)
Dv(60)	53.4553	72.3098	114.6689	μm	Dv(60)
Dv(70)	63.7003	88.4150	147.3606	μm	Dv(70)
Dv(80)	92.0202	120.4006	189.5668	μm	Dv(80)
Dv(90)	111.1426	153.1794	258.8265	μm	Dv(90)
Filling set position	5.20	5.92	6.95	mm	FSP
Main compression set Position	5.47	5.93	6.56	mm	MCSPo
Production speed	14.96	18.01	19.03	10000 tablets/h	PS
Rail Temperature	2.540	34.215	194.610	°C	RT
Average tablet weight	0.778	0.802	0.823	g	ATW
Average tablet hardness	80.0978	107.376	119.974	N	ATH

**Table 2 pharmaceutics-17-01510-t002:** Control performance evaluation index.

Evaluation Indicators	Formula
MSE	$MSE=\frac{1}{N}\sum_{n}^{i=1}{\left(y(i)-\hat{y}(i)\right)}^{2}$
IAE	$IAE=\sum_{i=1}^{N}\left|y_{d}(i)-y(i)\right|$
ITAE	$ITAE=\sum_{i=1}^{N}\left|y_{d}(i)-y(i)\right|\times i$

**Table 3 pharmaceutics-17-01510-t003:** Principal component loadings matrix.

Particle Size	PC1	PC2
Dv(10)	0.252	0.385
Dv(20)	0.261	0.321
Dv(30)	0.249	0.156
Dv(40)	0.265	0.082
Dv(50)	0.258	0.023
Dv(60)	0.263	−0.057
Dv(70)	0.255	−0.183
Dv(80)	0.246	−0.316
Dv(90)	0.269	−0.405

**Table 4 pharmaceutics-17-01510-t004:** Evaluation indexes of IL-MPC under different batch conditions.

	Iterations	3rd Iteration	7th Iteration	10th Iteration	15th Iteration	18th Iteration
MSE	Weight (g2)	6.55698 × 10^−4^	1.4287 × 10^−4^	8.0393 × 10^−5^	2.2121 × 10^−5^	3.2448 × 10^−6^
	Hardness (N2)	1.4331 × 10^3^	508.1883	59.0167	13.1855	6.1626
IAE	Weight (g)	3.0105	1.3826	1.0211	0.5007	0.1593
	Hardness (N)	4.4397 × 10^3^	2.6338 × 10^3^	850.7811	373.9173	184.2277
ITAE	Weight (g∙s)	170.1868	75.8181	53.6621	23.6157	6.1641
	Hardness (N∙s)	2.8363 × 10^5^	1.6811 × 10^5^	5.3681 × 10^4^	2.2641 × 10^4^	6.7289 × 10^3^

## Data Availability

The original contributions presented in this study are included in the article. Further inquiries can be directed to the corresponding author.

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
