# Peer review of "Adaptive Closed-Loop Control System for the Optimization of Tablet Manufacturing Processes"

_pharmaceutics, 2025, doi:10.3390/pharmaceutics17121510_

Round 1
Reviewer 1 Report
Comments and Suggestions for Authors
The authors presented an integrated ILC and MPC in a pharmaceutical tableting process. Historical process data was used to develop the process model in MPC. Process can be subject to variation or uncertainties from many aspects and also at different time scale. Within a batch variations can be handled by real-time feedback control with control actions in seconds or minutes, e.g., PID or MPC; while, variations from batch to batch can be compensated in ILC with a time scale of data to day. However, the proposed IL-MPC strategy is not clearly stated, see Eqs. (1) and (4). The authors seem not very familiar with a real process control and automation system on the floor, i.e., the fundamental hierarchical servo PLC, drive motor, actuators in a tablet machine.
Please see my comments below.
(1) On page 1, section 1. Introduction, there are many research efforts in the integrated iterative learning control or batch to batch with (nonlinear) model predictive control in pharmaceutical industry, e.g., crystallization. The authors should provide a short literature review in the introduction section and highlight the innovation of the presented method.
(2) On page 2, line 141 and 149, the authors shall provide more details of the multivariate prediction model. What kind of the model was adopted, ARMAX? Please also provide what model parameters are updated from batch to batch. And how the learning gain K matrix is designed.
(3) On page 11, Figure 4, the comparison of PID and MPC on in-batch control, there was not "significantly" reduced in control error range. Please clarify. The parameters in the Gaussian disturbances of Eq. (9) are marginal (0.015). What are the real variance in the process? Any estimate from the Table 1.
(4) On page 12, Figure 5, it is not realistic for the process to take 18 iteration to reach the target to produce on-spec product. The in-batch control by PID or MPC should correct the in-batch variations or uncertainties. The authors should provide the trajectories of the model parameters updated throughout the iterations.
(5) On page 13 and 14, Figure 6 and Figure 7 horizontal axial of "Time (s)" were not possible. The tablet sampling time was 2 min. And how did the authors handle the time delay in the 10 tablets analysis time for weight and hardness. The hardness measurement is destructive. It will causes a lot of waste for a 2 min sampling time.
Author Response
We have meticulously addressed each of your comments and implemented the corresponding revisions, as detailed in the attached document.

Reviewer 2 Report
Comments and Suggestions for Authors
The work is generally well written, interesting and seems to be carried out properly.
One general question remains after reading the manuscript. The authors compare IL-MPC to PID, and this is relevant, but do not clearly compare the performances with other MPC approaches. This is missing and would greatly improve the manuscript.
Figure 1. You state that IL-MPC is structured as a typical in which the ILC is presented as a simple constraint. The structure presented is a very typical one on the surface. As such, the novelty is not obvious. However, you go on to list a number of advantages (lines 114-124) without substantiating these claims. This has to be addressed.
Section 4.1 “The material attributes consisted mainly of moisture content and particle size characteristics: moisture was monitored online in real time by microwave sensors, while particle size was determined by a Malvern laser particle sizer (Table 1).” Provide details on these measurements. How they were obtained.
Section 4.1 You mention particle size. This will be a distribution. How was it treated in the context of IL-MPC?
Section 5: “principal component particle size (PC1 and PC2) exhibit a certain degree of correlation with the target variables”. Please rephrase this as the components must be orthogonal.
Author Response

(The authors gave the same response as above.)

Reviewer 3 Report
Comments and Suggestions for Authors
Introduction:
- lines 43-46: The statement could not be generalized, because of the success of thousands of tablet formulations at the level of industry and R&D. Authors should present the cases of such fluctuations.
- Lines 53-54: "The existing 52 strategies can be broadly categorized into two types: model-free control and model-based 53 control" For the readers, please define each mode!
- Lines 63-66: what is the difference between MPC, PID and QbD approach.
Does QbD fail to predict the tablets attributes based on CMAs and CPPs? - Lines 74-81: Authors should present industrial and (R&D) examples showing tthe applications of these strategies, and thier impacts on batch-to batch consistency. Basic theory of iterative learning predictive control 100: 1. Lines 105-108: Give examples and references! 2. Lines 111-113 is Repetitive 3.line 115: "The controller utilizes the error data from historical batches for integration and applies the accumulated error as a correction to the current batch. This iter-116 ative learning and compensation process enables the system to continuously optimize over repeated operations, significantly eliminating repetitive errors between batches, thus improving the consistency of the entire production process and product quality. In addi-tion, the ILC strategy enhances the robustness" add references.Line 123: why the authors focus the tablet weight and hardness even though these attributes are not superior for tablet evaluations? 4.line 123: why the authors focus the tablet weight and hardness even though these attributes are not superior for tablet evaluations??? 5. Equation 1: Reference for the equation. 3. 3.MPC-ILC control strategy for tablet quality assurance 1.Lines 157-164: Add References. 4. materials and method: Experimental Design 1. Details of tablet composition and compression tooling such as punch specifications, diameter, etc..should be described.
Author Response

(The authors gave the same response as above.)

Reviewer 4 Report
Comments and Suggestions for Authors
The manuscript deals with an iterative formulation and process development for a ternary blend for roller compaction/dry granulation process. The optimal formulation space was defined for a fixed process parameter setting and for pre-defined target values for response variables bulk density and angle of repose of the granulated product and breaking force of the tableted product.
I recommend the current status of the manuscript for acceptance after minor revision. The authors are advised to address the following minor comments and suggestions:
- The title of the article can be improved to better consider the purpose of the work, e.g. Adaptive closed-loop control system for the optimization of tablet manufacturing processes
- Throughout the manuscript “et al” should be corrected to “et al.”.
- The formatting should be checked and adapted according to the journal guidelines, e.g. Figure and Tablet in bold, tables should ideally not be divided across pages.
- Line 164: Please check wording and adapt if needed.
- Line 179: Please check the formula and adapt if needed.
- Line 279: I would suggest transferring the table into a figure.
- Line 294: A title should be added to the y-axis “Spearman correlation coefficient”.
- Line 315: Please add an explanation for sampling time (k).
Author Response

(The authors gave the same response as above.)

Round 2
Reviewer 1 Report
Comments and Suggestions for Authors
Thank you for addressing the comments. How about the physical implementation of the proposed control strategy into the tablet machine? Since the results shown here are based on simulations with noise of zero mean, please also clarify. Any challenges?
Pharmaceutical tableting process is a fast unit operation with expensive drug substance and in a regulated GMP environment. The proposed method shows 18 iterations to converge, which means a lot of out-of-spec materials will be produced or should be diverted. Please comment.
Author Response
The revised content is attached.

Reviewer 2 Report
Comments and Suggestions for Authors
I have no further comments.
Author Response
Thank you for your positive feedback. The revised content is attached.

Reviewer 3 Report
Comments and Suggestions for Authors
The authors responded to the comments, and the manuscript is now ready to be accepted.
Author Response
Thank you for your positive feedback.